# Mollification of Doxorubicin (DOX)-Mediated Cardiotoxicity Using Conjugated Chitosan Nanoparticles with Supplementation of Propionic Acid

**DOI:** 10.3390/nano12030502

**Published:** 2022-01-31

**Authors:** Durairaj Siva, Subramanian Abinaya, Durairaj Rajesh, Govindaraju Archunan, Parasuraman Padmanabhan, Balázs Gulyás, Shanmugam Achiraman

**Affiliations:** 1Department of Environmental Biotechnology, Bharathidasan University, Tiruchirappalli 620 024, Tamil Nadu, India; sivabio88@gmail.com (D.S.); abinayaebt@gmail.com (S.A.); 2PG and Research Department of Biotechnology, Srimad Andavan Arts & Science College (Autonomous), Tiruchirappalli 620 005, Tamil Nadu, India; 3Department of Biotechnology, Cauvery College for Women (Autonomous), Tiruchirappalli 620 018, Tamil Nadu, India; 4Research Institute in Semiochemistry and Applied Ethology (IRSEA), Quartier Salignan, 84400 Apt, France; r.durairaj@group-irsea.com; 5Department of Animal Science, Bharathidasan University, Tiruchirappalli 620 024, Tamil Nadu, India; garchu56@yahoo.co.in; 6Dean of Research, Marudupandiyar College, Vallam, Thanjavur 613 403, Tamil Nadu, India; 7Lee Kong Chian School of Medicine, Nanyang Technological University, Singapore 636921, Singapore; 8Cognitive Neuroimaging Centre, 59 Nanyang Drive, Nanyang Technological University, Singapore 636921, Singapore; 9Department of Clinical Neuroscience, Karolinska Institute, 17176 Stockholm, Sweden; balazs.gulyas@ntu.edu.sg

**Keywords:** doxorubicin, PPARγ, chitosan nanoparticles, cardiotoxicity, propionic acid

## Abstract

Doxorubicin is an extensively prescribed antineoplastic agent. It is also known for adverse effects, among which cardiotoxicity tops the list. The possible mechanism underlying doxorubicin (DOX)-mediated cardiotoxicity has been investigated in this study. Further, to reduce the DOX-mediated cardiotoxicity, DOX was conjugated with Chitosan Nanoparticles (DCNPs) and supplemented with propionic acid. Initially, the drug loading efficacy and conjugation of DOX with chitosan was confirmed by UV–Visible Spectroscopy (UV) and Fourier Transform Infrared Spectroscopy (FTIR). The average sizes of the synthesized Chitosan Nanoparticles (CNPs) and DCNPs were measured by Dynamic Light Scattering (DLS) analysis as 187.9 ± 1.05 nm and 277.3 ± 8.15 nm, respectively, and the zeta potential values were recorded as 55.2 ± 0.7 mV and 51.9 ± 1.0 mV, respectively. The size and shape of CNPs and DCNPs were recorded using a High-Resolution Electron Microscopy (HRTEM). The particles measured <30 nm and 33–84 nm, respectively. The toxic effects of DCNPs and propionic acid were evaluated in rat model. The data from the electrocardiogram (ECG), cardiac biomarkers, Peroxisome proliferator-activated receptor gamma (PPARγ) and histological observations indicated evidence of DOX-mediated cardiotoxicity, whereas the administration of DCNPs, as well as Propionic Acid (PA), brought about a restoration to normalcy and offered protection in the context of DOX-induced cardiotoxicity.

## 1. Introduction

Doxorubicin hydrochloride is an effective antineoplastic drug that helps mitigating various types of cancers [1]. Although effective, the long-term use of this drug causes severe side effects, which limits its use. Cardiotoxicity is one of the dreadful side effects caused by DOX [2,3]. DOX-mediated toxicity may be revealed within months or years after the completion of treatment [4]. The principal aim of the present study has been to explore the possible mechanism underlying DOX-mediated cardiotoxicity and to enhance the applicability of the drug by reducing its toxicity. Arunachalam et al. [5] predicted that DOX treatment might alter glucose and lipid profiles in circulation, which might alter adipogenesis in adipocytes and mimic type 2 diabetes [5]. They further hypothesized that the administration of Doxorubicin might affect the physiology of adipocytes by disturbing the expression of PPARγ. PPARγ acts to enhance the cellular uptake of glucose through adipose tissue remodeling, accompanied by diminished free fatty acids (FFA) and greater adiponectin release, which activates adenosine mono-phosphate kinase (AMPK) [6]. AMPK has been suggested to be responsible for the uptake of glucose by activating glucose transporter type 4 (GLUT4) vesicles. Published literature confirms that PPARγ directly regulates the expression of fatty acid transport protein (FATP)/cluster of differentiation 36 (CD36), which is responsible for the cellular uptake of long-chain fatty acids [5]. The downregulation of PPARγ could possibly affect lipid and glucose metabolism directly by inhibiting the uptake of glucose and fatty acids into the cells. To reduce DOX-mediated toxicity, two different novel aspects were tested in this study, namely: conjugation of chitosan nanoparticles with DOX (DCNPs) that enhances drug delivery during treatment, as has happened in the case of conjugation of DOX with hyaluronic acid [3], and multifunctional dendrimers [7]; and secondly, the reduction of DOX-mediated toxicity by supplementation of propionic acid (PA).

Chitosan was chosen in this study, as it is a highly practiced biopolymer for drug delivery in light of its desirable properties, such as biocompatibility, nontoxic nature and easy biodegradability [8]. Chitosan, a deacetylated derivative of chitin, is a hydrophilic cationic polysaccharide and a major component of crustacean shells [9]. It is a de-N-acetylated analog of chitin, composed of linear glucosamine residues. This polymer can be depolymerized chemically, physically or by means of enzymatic methods. D-glucosamine derivatives have long attracted attention in view of their hemostatic and antitumor properties [10]. Due to its advantageous biological properties, chitosan has gained much attention in various industrial applications, such as biomedical and pharmaceutical industries [11], food industries [12], agricultural practices [13], water purification [14] and paper production [15]. Data from studies conducted earlier show that CNPs are effective in reducing the serum lipid levels in rats fed a high-fat diet and show efficient hypolipidemic activity [16] and anticancer activity [17]. Chitosan upregulates the activity of AMPK and enhances energy metabolism [18]. Dietary chitosan has been shown to enhance blood lipid profiles in patients with diabetic and renal diseases. CNPs are known for their beneficial lipid-lowering effect on plasma cholesterol, which may play an important role in prevention and treatment of cardiovascular diseases [19]. 

PA is a short-chain fatty acid that can activate PPARγ. Therefore, we hypothesized that propionic acid, a PPARγ agonist, has the ability to reduce fatty acid levels in plasma via inhibition of lipolysis and induction of lipogenesis in adipose tissue; such incidents may reduce the cardiotoxicity caused by DOX [20]. Considering the significant therapeutic use of DOX in the clinical scenario, the present study was designed to characterize the synthesized nanoparticles (CNPs and DCNPs) and subject them to evaluation of their efficacy to alleviate cardiotoxicity. Further, the effect of PA supplementation along with DCNPs was analyzed at the molecular level. Similar to CNPs, PA has already been reported for its lipid-lowering ability [20].

## 2. Experimental Section

### 2.1. Materials

Low molecular weight chitosan and sodium tripolyphosphate were purchased from Sigma-Aldrich. Acetic acid (glacial) 100% and glycerol were purchased from Merck. The drug doxorubicin hydrochloride (ADRIM) was purchased from Fresenius Kabi Oncology Ltd. (Baddi, Himachal Pradesh, India). Ultrapure Millipore water was obtained in a Milli-Q^®^ Advantage A10 Water Purification System. All chemicals and reagents involved in the experiments were used without any further purification. C-reactive protein (CRP) was purchased from BioSystems, Spain. Creatine kinase–MB (CKMB) and lactate dehydrogenase (LDH) kits were purchased from Agappe Diagnostics Limited, India.

### 2.2. Preparation of Chitosan Nanoparticles (CNPs) and DOX-Conjugated Chitosan Nanoparticles (DCNPs)

CNPs were synthesized according to the modified method of Calvo et al., based on the ionotropic gelation of chitosan with sodium tripolyphosphate anions [21], with modifications adopted from López-León et al. [9]. The nanoparticles were obtained upon the addition of 1.2 mL (0.84 mg/mL, *w*/*v*) of sodium tripolyphosphate aqueous basic solution to 3 mL of the chitosan acidic solution (2 mg/mL, *w*/*v*) under magnetic stirring at room temperature to obtain a homogenous solution. The nanoparticles were concentrated by centrifugation at 16,000× *g* in a 10-μL glycerol bed for 30 min. The centrifugation and resuspension in Milli-Q water were repeated thrice to ensure the effective separation of CNPs from free entities. DCNPs were also synthesized using the same protocol as adopted for the synthesis of CNPs. However, here, DOX was added at varying concentrations in 3 mL of chitosan acidic solution (2 mg/mL, *w*/*v*) and mixed well under magnetic stirring at room temperature. Later, in a similar protocol, 1.2 mL (0.84 mg/mL, *w*/*v*) of sodium tripolyphosphate aqueous basic solution was added and again allowed to mix well under magnetic stirring at room temperature to obtain a similar pattern of solution. DCNPs were also centrifuged at 16,000× *g* in a 10-μL glycerol bed for 30 min in the same manner as for the synthesis of CNPs. The supernatant was collected to calculate the drug loading efficiency. During the conjugation and synthesis of DCNPs, the setup was kept in the dark to avert any photodegradation process.

### 2.3. Drug Loading Efficacy

The quantity of DOX conjugated with the CNPs was calculated by the method of Arulmozhi et al. [22]. Initially, the λ-max value of DOX (480 nm) was identified using UV–visible spectroscopy. The quantity of unconjugated DOX was calculated from the supernatant by analyzing its absorbance at 480 nm using UV–visible spectrophotometry (Ultrospec 2100 pro, Amersham Biosciences, Amersham, Buckinghamshire, UK). The concentration of DOX was measured using a standard plot, and the percentage of conjugation was determined using the equation
$$\mathrm{Drug}\;\mathrm{loading}\;\mathrm{efficiency}=\frac{\mathrm{Total}\;\mathrm{quantity}\;\mathrm{of}\;\mathrm{DOX}-\mathrm{Unloaded}\;\mathrm{DOX}\;\mathrm{in}\;\mathrm{supernatant}}{\mathrm{Total}\;\mathrm{quantity}\;\mathrm{of}\;\mathrm{DOX}}\times 100$$

The most efficient DOX-conjugated CNPs was taken for characterization and further studies.

### 2.4. Characterization Studies

The synthesized CNPs and DCNPs were stored at 4 °C. Morphological features of the nanoparticles were studied in a High-Resolution Transmission Electron Microscope (HR-TEM), which provides an initial confirmation that the synthesized nanoparticles are in the nano range [23]. The average size, ζ-potential and dispersity index of the nanoparticles were analyzed by Malvern Zetasizer Nanoseries compact spectrophotometer (Malvern Instruments Ltd., Malvern, UK). The analysis was performed in triplicate to obtain statistically significant data [24]. Finally, DOX, CNPs and DCNPs were analyzed by Fourier Transform Infrared spectroscopy (FTIR) to validate the available functional groups. A Spectrum RX1 instrument provided with diffuse reflectance mode at 4 cm^−1^ resolution of KBr pellets was used. The spectra were read at the wavelength range of 400 nm^−1^ to 4000 nm^−1^ [25].

### 2.5. Experimental Animals

Male albino Wistar rats (*Rattus norvegicus*), weighing 230 ± 20 g, were purchased from Kerala Veterinary and Animal Sciences University, Trissur, Kerala, India. The animals were maintained in polypropylene cages at 25 ± 2 °C and 45–55% humidity with 12 h light/dark cycle. Animals were allowed free access to standard laboratory feed (Sai Enterprisei, Chennai, India) and water ad libitum. The Institutional Animal Ethical Committee at Bharathidasan University had approved the experiments (BDU/IAEC/2015/NE/19/Dt.17.03.2015).

### 2.6. Experimental Design

After 10 days of laboratory acclimation, the experimental animals (35 rats) were randomly divided into seven groups: Group 1 served as intact; Group 2 (DOX) was administered Doxorubicin at a cumulative dose (CD) of 10 mg/kg body weight; Group 3 was administered chitosan nanoparticles (CNPs) intraperitoneally at a dose of 12 g/kg body weight CD; Group 4 received DOX-conjugated chitosan nanoparticles (DCNPs) at a dose of 12 g/kg body weight cumulative dose (CD); Group 5 was provided with propionic acid (PA) at a dose of 100 mg/kg body weight through oral route as CD; Group 6 was DOX + PA-treated and received both DOX and PA concurrently at doses of 10 mg/kg body weight CD and 100 mg/kg body weight CD, respectively; Group 7 was DCNPs + PA-treated, where the rats were given 12 g/kg body weight (CD) of DCNPs and 100 mg/kg body weight (CD) of PA concurrently. The doses were equally split and administered for 30 days [26]. The body weight was monitored daily during the experimental period. At the end of the experiment, the ECG was measured. Blood was collected by orbital bleeding. Serum was separated by centrifugation at 2500 rpm for 15 min and stored at −20 °C for conducting biochemical analysis. Heart and aorta were dissected out from the animals after sacrifice under euthanasia and used for histological analysis and biochemical, as well as antioxidant, assays. 

### 2.7. Evaluation of ECG Alterations

Twenty-five days after commencement of the experiment, the animals were anesthetized by intraperitoneal injection of ketamine hydrochloride and xylazine cocktail (ketamine hydrochloride 50 mg/kg; xylazine 5 mg/kg) and kept in supine position. The electrocardiogram (MP 100, Biosystems, Inc., Cumming, GA, USA) was used to record the classic limb leads (right arm, left arm and left leg, respectively, for aVR, aVL and aVF) at 50 mm/s and 20 mV/mm. The amplitude of the traces between the peaks (QRS value) and intervals of ECGs, such as PR-, ST- and QT, and heartbeat rate, were recorded using data acquisition software (Acknowledge 3.9.0, Goleta, CA, USA).

### 2.8. Biochemical and Antioxidant Analyses

The functional efficiency of the heart was assessed by analyzing biochemical markers, such as C-reactive protein (CRP), creatine kinase–MB (CKMB), lactate dehydrogenase (LDH) and LDL cholesterol in the serum of experimental animals using a semi-analyzer (BioSystems, BTS350). Enzymatic antioxidants, such as superoxide dismutase (SOD), glutathione-S-transferase (GST) and lipid peroxidation marker malondialdehyde (MDA), were measured in the heart tissue. The level of SOD was determined by measuring the inhibition of auto-oxidation of pyrogallol, as mentioned in the method of Marklund and Marklund [27]. Glutathione-S-transferase (GST) was measured according to the protocol of Habig et al. [28]. The amount of MDA produced was recorded adopting the method of Ohkawa et al. [29].

### 2.9. RNA Isolation and qPCR Analysis

Total RNA was extracted from the heart tissue using TRIzol RNA-ase reagent kit (EZ one step RNA reagent, Cat. No.BS410A, Biobasics., Markham, ON, Canada), as prescribed in manufacturer’s instruction. The concentration and purity of RNA were measured using a Biophotometer (Biophotometer Plus, Eppendorf, Hamburg, Germany). The isolated RNA was used for the expression study of PPARγ level in the heart tissue. Two micrograms of total RNA was reverse-transcribed into cDNA using Verso cDNA synthesis kit (Cat. No-#AB-1453/B Thermo Scientific, Waltham, MA, USA) in a Bio-Rad S1000TM thermal cycler (thermal cycler with dual 48/48 fast reaction module, Bio-Rad Laboratories Inc., Hercules, CA, USA). Real-time PCR was carried out as follows: 72 °C for 5 min, 4 °C for 10 min, 42 °C for 60 min, 72 °C for 10 min and 92 °C for 2 min. Real-time PCR was performed in a total volume of 20 μL, with 10 μL of 2× SYBR Premix Ex Taq, Tli RNaseH Plus (Takara Bio Inc., Cat.#RR420A), 1.0 μL of each primer (3 pmoles each), 2.0 μL of cDNA and 6.0 μL sterile H_2_O, and was conducted in triplicate for each sample in a Roche light cycler 96 L600 Real-Time PCR system. Primers were designed manually using the sequence within the coding domain, adopting the online software oligo-analyzer, reverse complement and similarity searches in BLAST. The primers were purchased from Shrimpex Biotech Services Pvt. Ltd. (Chennai, India). 18S rRNA and β-actin were used as the endogenous references to normalize gene expression, and 2^−ΔΔCT^ method was used to determine the differences in mRNA content between the samples [26]. 

Details of the primers and the size of their amplification product are as follows:

Gene    Sequence        Desired size

PPARγ   F 5′- GCC AAG GCG AGG GCG ATC-3′    180 bp

PPARγ   R 5′- CAC GGA TCG AAA CTG GCA CC- 3′

### 2.10. Histological Examination

Dissected heart tissue was immediately washed thoroughly in normal saline, in order to remove the blood clots/stains, and fixed in 10% formalin. Paraffin sections, 5 µm thick, were obtained and stained with hematoxylin and eosin. Histological changes were observed in a light microscope and recorded using the Magnus Microscope Image Projection System (Magnus Analytics, Mathura Road, New Delhi, India).

### 2.11. Statistical Analysis

The values were expressed as mean ± SD. Statistical significance was calculated by one-way analysis of variance (ANOVA) using SPSS version 25 (IBM Corp, New York, NY, USA), and individual comparisons were obtained by Duncan’s Multiple Range Test [30]. A *p* value of <0.05 was considered as indicative of significant differences between groups.

## 3. Results and Discussion

### 3.1. Characterization of CNPs and DCNPs

In light of the fact that CNPs are good nanocarriers for intracellular drug delivery, it was hypothesized that they may reduce the toxicity caused by DOX. Thus, in the present study, DOX-conjugated chitosan nanoparticles were synthesized and tested in the context of cardiotoxicity in an animal model [31]. Initially, a conjugation of DOX with CNPs was confirmed visually from the red color of the pellet obtained from centrifugation. After conjugating DOX at varying concentrations (0.05, 0.1, 0.5, 1, 2 and 3 nM) with the CNPs, the drug loading efficiency and the optimum concentration of the drug were measured using UV–visible spectrophotometry at 480 nm (Appendix A). From the analysis, it was confirmed that the drug at the highest concentration (3 mM) tested had the highest drug loading efficiency. Hence, this concentration was chosen for further characterization studies. The nature of interaction between the drug and CNPs was established using FTIR spectroscopy (Figure 1). 

The similarities and changes or shifts in the formation of peaks help in understanding the chemical interactions, thereby providing confirmation for the conjugation of the drug with nanoparticles [32,33]. After the confirmation of DOX conjugation with CNPs, the synthesized CNPs and DCNPs were studied by Zeta sizer analysis in order to confirm the presence of nanoparticles, their average particle size and their dispersion index (Figure 2a,b).

Although the average size of the nanoparticles was beyond the nano range, the average size of the synthesized CNPs and DCNPs being 187.9 ± 1.05 nm and 277.3 ± 8.15 nm, respectively, most of the nanoparticles were in less than 100 nm size range, which was further confirmed by the size distribution analysis by number (Appendix A). The polydispersity indices (PdI) of CNPs and DCNPs were 0.212 and 0.215, respectively, which means the synthesized nanoparticles were monodispersed. Particle size and its distribution are the basic factors that influence the in vivo distribution, activity in the biological system, toxicity and targeted delivery of drugs conjugated with the nanoparticles [22].

To confirm the stability, the samples were subjected to zeta potential analysis using the same instrumentation as above (Figure 2c,d). The stability of the nanoparticles was determined by the zeta potential caused by the available net electrical charge within the area limited by the slipping plane and also depended on the site of that plane. The zeta potential of the synthesized CNPs and DCNPs were 55.2 ± 0.7 and 51.9 ± 1.0, respectively. Between CNPs and DCNPs, the latter showed a slight increase in the average particle size and a slight decrease in its stability. The observed changes in the DCNPs might be due to the reactivity of the functional groups present in the drug conjugated with nanoparticles [22].

Though the average particle size has been confirmed using DLS analysis, a few large-sized particles were detected, which may be due to agglomeration or contamination causing imprecisions during the analysis. Hence, it became essential to characterize the size and shape of the nanoparticles through HRTEM analysis [34]. It was confirmed that all the synthesized CNPs were less than 20 nm in size and irregular spherical in shape (Figure 3a–c).

In agreement with the DLS analysis, it was clear that the nanoparticles were arranged in a monodispersed pattern. The HRTEM results of DCNPs showed that the DCNPs were in the size range 33—84 nm and irregular spherical in shape (Figure 3d–f). In contrast to their PdI value, as derived from the zeta potential analysis, the particles showed aggregation with one another and were polydispersed.

### 3.2. Changes in Body Weight

The physiological and cardiac parameters were studied in the Wistar rats, which were administered the synthesized CNPs and DCNPs and compared with the DOX-administered rats. During the study period, the body weight changes were monitored to evaluate the stress conditions and health of the experimental animals. The changes in the body weight are presented in Table 1. 

Animals in all experimental groups recorded an increase in body weight at the end of the experiment. It reveals age-related growth of the animals. However, the range of body weight increase was interrupted or reduced in DOX and DOX + PA groups compared to intact and other treated groups. These changes may be caused by the effects of fear, hunger or thirst, which the rats experienced during the study period [35]. 

### 3.3. Electrocardiography

The ECG pattern reflecting electrical activity was analyzed at the end of the study (Figure 4). The waves produced by the electrophysiological pattern facilitate a diagnosis of ill health of the heart, if any. The ECG of experimental rats, depicted in (Figure 4), elaborates the ECG patterns, QRS, QT, ST and PR peaks, respectively. 

The heart of untreated control animals exhibited normal wave pattern. On the other hand, the DOX-administered animals had immature P peaks. The lack or alteration of the P waves in the ECG pattern is clear evidence of various cardiac arrhythmias. More specifically, the atrial fibrillation in rats is diagnosed by a lack of or immature P wave [36,37]. This abnormality was not observed in the animals administered with CNPs, DCNPs, PA, DOX + PA and DCNPs + PA. The ECG results clearly indicate an irregular functioning of the heart in DOX-administered rats.

### 3.4. Analysis of Cardiac Biomarkers

Apart from ECG, the level of cardiac biomarkers, such as CRP, CK-MB, LDH and LDL cholesterol, were evaluated in the serum of experimental rats (Figure 5). CRP is a common inflammation marker. Its level in the serum increases within a few hours of an onset of inflammation. 

Physicians believe that CRP is a stronger predictor of cardiac risk than LDL cholesterol [38]. The evaluation of CK-MB is a confirmatory analysis of cardiotoxicity after CRP test. Myocardium is the only major source of the CK-MB enzyme. Physicians recommend checking the level of this enzyme to diagnose disorders, such as myocardial infarction (heart attack), rhabdomyolysis (severe muscle damage), muscular dystrophy and acute kidney injury [39]. Elevation of CK-MB in serum is an obvious sign of myocardial damage. In the present study, elevated levels of CRP and CK-MB were recorded in the serum of DOX-administered rats compared to rats in other experimental groups, as has been reported by Cota et al. [40]. In fact, the serum of rats administered with CNPs, PA and DCNPs exhibited lower level of CRP and CK-MB [41]. Kim et al. reported that the treatment with anticancer drugs causes severe tissue injury, thereby increasing the level of LDH in the blood [42]. The level of LDH was elevated in the DOX-administered animals compared to animals in the other experimental groups, which confirms tissue injury [43]. Sharifinasab et al. reported that the administration of CNPs can regulate the level of LDH [44]. Similarly, reduced levels of LDH were produced in the serum compared to DOX-administered rats upon the administration of CNPs and DCNPs (Figure 5).

In the regulation of cholesterol in the liver, LDL cholesterol plays a major role in transporting cholesterol from liver to various peripheral tissues. Figure 5 shows the levels of LDL cholesterol in the experimental animals. Higher level of LDL cholesterol is closely connected with increase in the chances of developing cardiovascular disease. The level of LDL cholesterol in the present study was higher in DOX-administered rats than in the rats in other experimental groups. Similarly, the level of LDL and LDH was increased in DCNPs + PA [45]. The LDL cholesterol level recorded in DOX-treated rats was restored to normal in DCNPs-treated rats. The treatment with DOX + PA resulted in near-normal LDL cholesterol [46,47].

### 3.5. Antioxidants Assays

Further, to reflect on the status of cardiotoxicity, the heart tissue of experimental rats was subjected to analysis of antioxidants (SOD and GST) and lipid peroxidation marker (MDA) in order to find the oxidative stress (Table 1). Antioxidants are the vital natural defense system of the body against free-radical-induced tissue damage [48]. The level of cellular antioxidants, as found in this study, confirms the status of oxidative stress-mediated tissue damage [49]. In the present study, the heart tissue recorded low levels of SOD and GST in DOX- and DCNPs + PA administered rats compared to untreated control [50]. Rats in DCNPs and DOX + PA groups showed a slight increase in the level of SOD and GST compared to DOX-administered rats. Lipid peroxidation marker MDA was found to be increased in the DOX-administered group compared to the untreated control and other experimental groups [43]. Rats in DCNPs exhibited lower level of MDA compared to DOX-administered group, whereas CNPs and DOX + PA groups showed very low level of MDA compared to untreated control and other experimental animals. 

### 3.6. Histological Observation

Further evidence of DOX-induced oxidative stress was demonstrated by histological sections of the heart and aorta. From the observation of sections of the heart, it was clear that DOX-administered rats showed disruptions of the cardiac tissue and abnormal architecture compared to rats in untreated control and other experimental groups. Among the treatment groups, DCNPs and DOX + PA showed commendable recovery from the deformations caused by DOX. CNPs- and PA-administered groups showed near-normal histoarchitecture of the heart. On the other hand, the DCNPs + PA group showed drastic tissue damage in the heart. Similarly, an increase in aortic wall thickening was observed in DOX, DOX + PA and DCNPs + PA-administered animals compared to untreated control and other experimental rats. DCNPs- and DOX + PA-treated groups showed lesser thickening in the aortic walls compared to untreated control. There was no indication of formation of plaques in the arteries of any of the experimental groups.

### 3.7. qPCR Analysis

Having substantiated cardiotoxicity of DOX from the results obtained by analysis of the major cardiac well-being parameters, the next query was to infer how DOX induces cardiotoxicity. This led to an exploration of the possible mechanism of DOX cardiotoxicity at the molecular level. According to an earlier view of Arunachalam et al., the administration of DOX to rats may lead to cardiotoxicity by downregulation of PPARγ in heart muscle [5]. Quantitative expression of PPARγ–mRNA in the heart tissues of experimental animals is represented in Figure 6. In the qPCR analysis, a lower expression ratio of PPARγ transcript was recorded in DOX-administered animals compared to untreated control and other experimental animals. 

The PA-administered rats exhibited higher expression of PPARγ in comparison with untreated control and other experimental groups. The finding in an earlier study, conducted by Sa’ad et al., showed that PA has the ability to upregulate PPARγ expression, since it is a remarkable PPARγ agonist [20]. Upregulation of PPARγ was seen in CNPs, DCNPs and DOX + PA groups compared to untreated control and DOX-administered groups. Thus, it was demonstrated that DCNPs and PA reduced the cardiotoxicity caused by DOX through the upregulation of PPARγ. At the molecular level, reduced expression of PPARγ-mRNA in the heart tissue of DOX-administered rats provides concrete evidence for dysregulation of lipid metabolism (Figure 7).

## 4. Conclusions

The outcome of the study provides substantial evidence for understanding the effect of DOX-mediated cardiotoxicity through downregulation of PPARγ in cardiac tissues. The latter would impair uptake of glucose and fatty acids, leading to energy demand in the heart, which is reflected in the oxidative stress marker MDA and antioxidants SOD and GST in the heart tissue. The tissue damage arising as such was confirmed by the LDH, CK-MB and CRP levels, along with histological observations. The PPARγ expression of DCNPs- and PA-administered rats was high, and the cardiotoxicity produced by DOX was greatly reduced after conjugation of DOX with CNPs and upon the supplementation of PA. Our hypothesis that propionic acid, a PPARγ agonist, has the ability to reduce fatty acid levels in plasma via inhibition of lipolysis and induction of lipogenesis in adipose tissue is greatly substantiated in this study. Moreover, the study opens an effective way of managing DOX-mediated cardiotoxicity, i.e., via the administration of DOX-conjugated chitosan nanoparticles. Thus, this study offers a novel platform in the field of nanoparticle-mediated targeted drug delivery.

## Figures and Tables

**Figure 1 nanomaterials-12-00502-f001:**
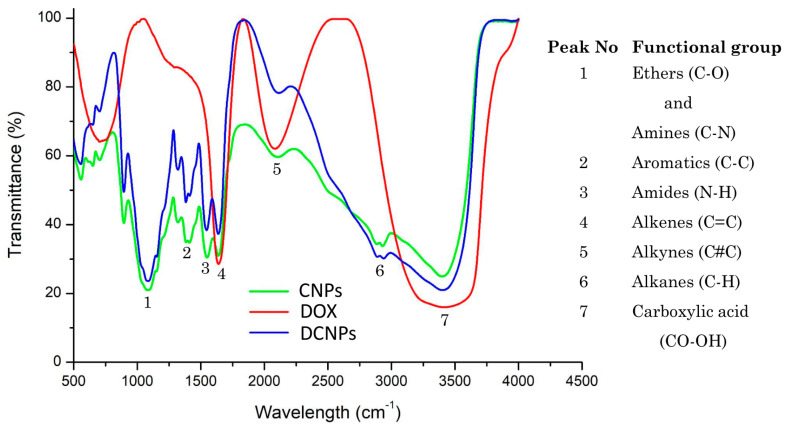
FTIR spectrum of CNPs DOX and DCNPs.

**Figure 2 nanomaterials-12-00502-f002:**
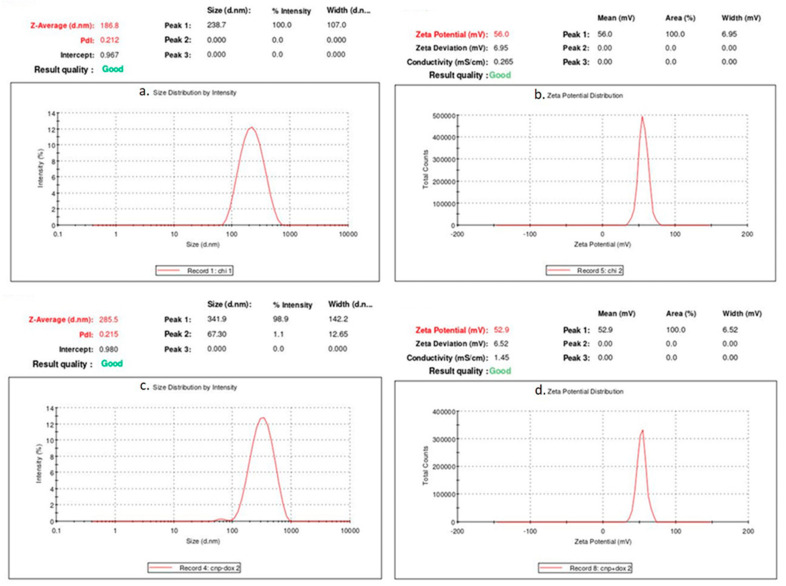
Dynamic Light Scattering and Zeta Potential Analysis; (**a**). Size Distribution of CNPs by Intensity, (**b**). Zeta Potential Distribution of CNPs, (**c**). Size Distribution of DCNPs by Intensity and (**d**). Zeta Potential Distribution of DCNPs.

**Figure 3 nanomaterials-12-00502-f003:**
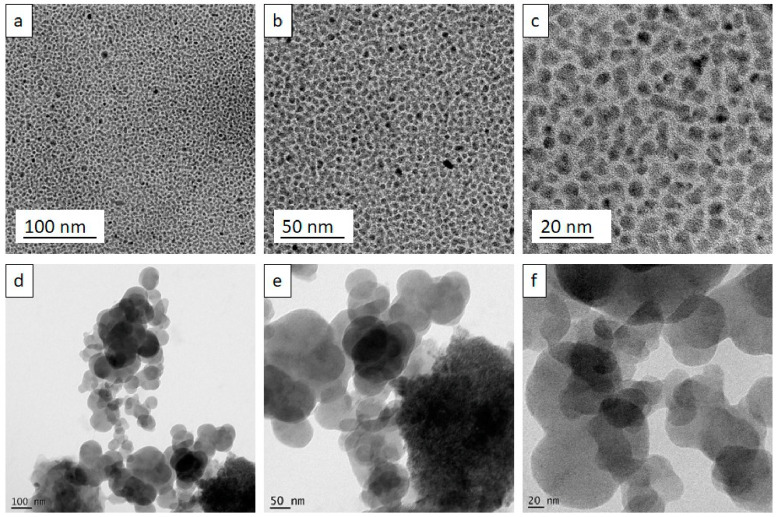
High-Resolution Transmission Electron Microscopical view of CNPs at (**a**). 100 nm, (**b**). 50 nm, (**c**). 20 nm and ACNPs at (**d**). 100 nm, (**e**). 50 nm, (**f**). 20 nm.

**Figure 4 nanomaterials-12-00502-f004:**
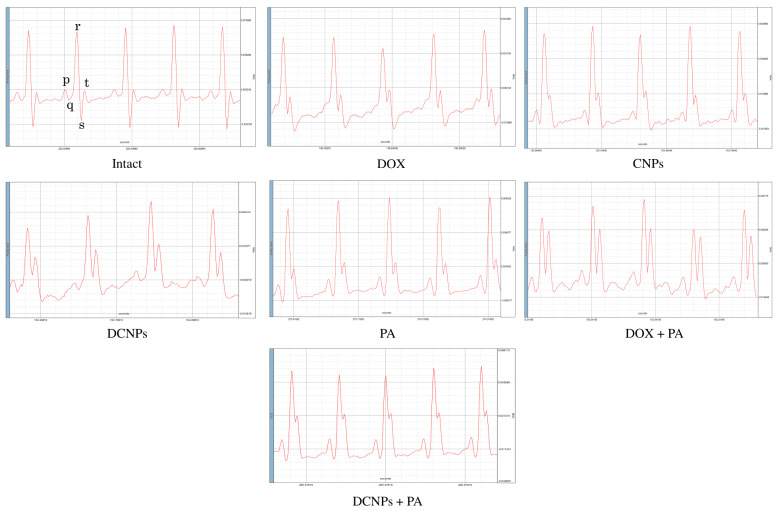
ECG Analysis of Experimental animals.

**Figure 5 nanomaterials-12-00502-f005:**
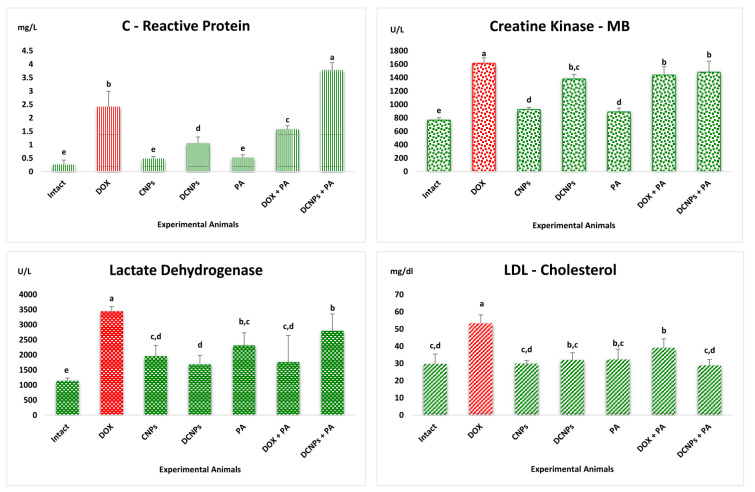
The effect of CNPs and PA on the level of Cardiac Biomarkers (C-Reactive Protein, Creatine Kinase–MB, Lactate Dehydrogenase and LDL Cholesterol) in the serum of experimental animals. Values were expressed as mean ± SD (*n* = 5). Bars with different alphabets are significantly different from each other, and bars with same alphabets have insignificant changes (*p* < 0.05).

**Figure 6 nanomaterials-12-00502-f006:**
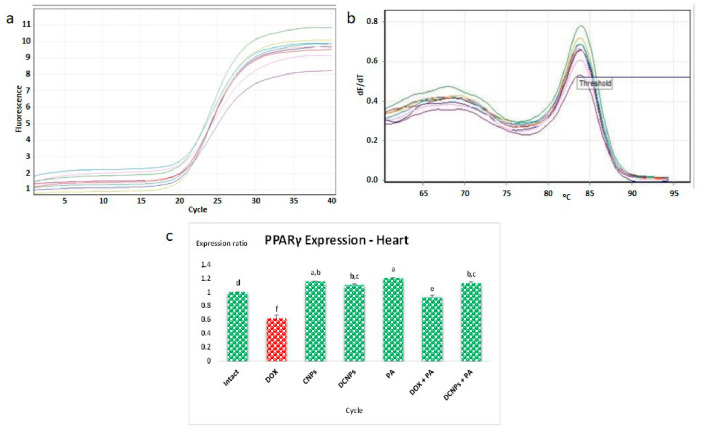
The expression level of PPARγ in Heart tissues of experimental animals. (**a**). Amplification curve (**b**). Melting curve and (**c**). Expression ratio. Values were expressed as mean ± SD (*n* = 3). Bars with different alphabets are significantly different from each other, and bars with same alphabets have insignificant changes (*p* < 0.05).

**Figure 7 nanomaterials-12-00502-f007:**
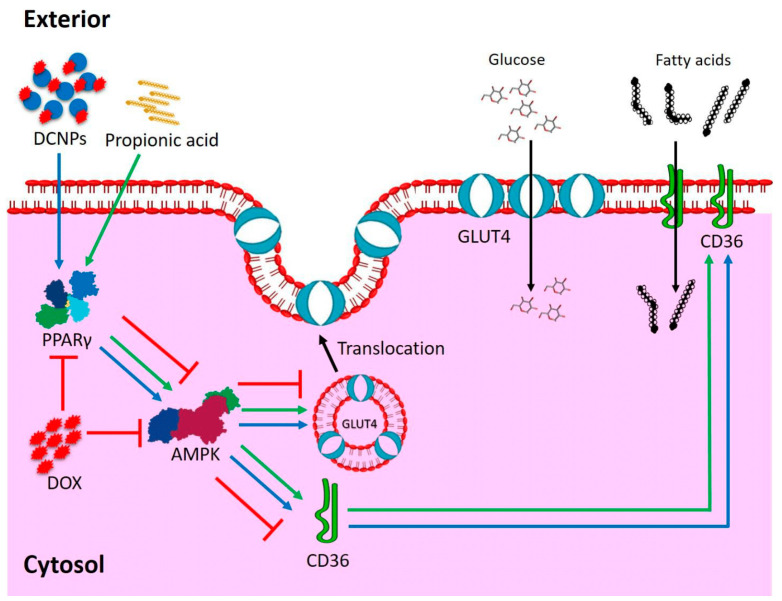
Mechanism—DOX-mediated Cardiotoxicity.

**Table 1 nanomaterials-12-00502-t001:** The effect of CNPs, PA and DCNPs on Body Weight, Heart Weight, Heart Beat and the level of antioxidants in Heart tissues of experimental animals. Values were expressed as mean ± SD (*n* = 5). Values with different alphabets are significantly different from each other, and values with same alphabets have insignificant changes (*p* < 0.05).

	Intact	DOX	CNPs	DCNPs	PA	DOX + PA	DCNPs + PA
Body Weight changes							
Initial (g)	161 ± 11 ^c^	162 ± 13 ^b,c^	178 ± 17 ^b^	173 ± 15 ^b,c^	160 ± 12 ^c^	160 ± 12 ^c^	176 ± 10 ^b,c^
Final (g)	254 ± 21 ^b,c^	227 ± 21 ^c,d^	250 ± 34 ^b,c,d^	290 ± 16 ^a^	265 ± 13 ^a,b^	224 ± 27 ^d^	245 ± 15 ^b,c,d^
Antioxidants							
Superoxide Dismutase (U/mg Protein)	14.7 ± 0.8 ^a^	5.5 ± 0.6 ^d^	10.9 ± 1.8 ^b^	8.0 ± 0.8 ^c^	11.9 ± 0.7 ^b^	7.9 ± 0.0 ^c^	6.1 ± 0.07 ^d^
Glutathione S Transferase (U/mg Protein)	0.043 ± 0.0 ^a^	0.011 ± 0.0 ^d^	0.032 ± 0.0 ^b^	0.035 ± 0.0 ^b^	0.031 ± 0.0 ^b^	0.024 ± 0.0 ^c^	0.033 ± 0.0 ^b^
Malondialdehyde (*n* mol/mg Protein)	0.86 ± 0.13 ^e^	3.02 ± 0.68 ^a^	0.62 ± 0.09 ^e^	1.56 ± 0.20 ^d^	0.56 ± 0.09 ^e^	2.03 ± 0.14 ^c^	0.54 ± 0.23 ^b^

## Data Availability

The data is available on reasonable request from the corresponding author.

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
