# Peer review of "Mollification of Doxorubicin (DOX)-Mediated Cardiotoxicity Using Conjugated Chitosan Nanoparticles with Supplementation of Propionic Acid"

_nanomaterials, 2022, doi:10.3390/nano12030502_

Round 1
Reviewer 1 Report
This work studied mollification of Doxorubicin (DOX) mediated cardiotoxicity using conjugated chitosan nanoparticles with supplementation of propionic acid. There are some concerns in this manuscript as follows:
- Abstract:
- The meaning of the abbreviations should be clearly defined at their first mention (e.g. UV in line 6 of the abstract).
- Line 13: Peroxisome proliferator-activated receptor gamma is NOT a specific cardiac marker. It is present in many tissues of the body other than the heart. Please, revise
- Line 16: “offer” should be replaced with “offered”.
- Key words: The meaning of “PPARγ” should be clearly mentioned.
- Introduction:
- The novel points in this study should be clarified because there are previous studies that demonstrated that propionic acidemia may lead to cardiomyopathy and this is against the hypothesis of the current study.
- The sentence “Similar to CNPs, PA has been already reported for its lipid lowering ability [20].” In the second paragraph should be transferred to the third paragraph .
- The suggested role of propionic acid in protection against DOX-cardiotoxicity should be re-written in a more-detailed manner to explain the hypothesis of the present study and to clarify the proposed mechanisms.
- The reference number [20] used for delineation of the role of propionic acid is old (2010). Please, use updated references.
- The meaning of the abbreviations should be clearly defined at their first mention (e.g. PPARɤ).
- Materials and methods:
- The exact source, concentrations and the catalogue numbers of the used drugs, kits and chemicals should be mentioned.
- A reference for the equation used for determination of “Drug loading efficacy” should be added.
- A reference for the methods used for “characterization studies” should be added.
- How did you know that the animals were acclimatized?
- The name of the institution that gave the Ethical approval should be mentioned.
- The number of animals used in this study should be mentioned.
- In experimental design line 3: Replace “Adriamycin” with “Doxorubicin”.
- References for the used doses of drugs and duration should be added.
- How the doses were split?
- The frequency of administration of DOX should be clarified.
- “The body weight was monitored regularly during the experimental period” Please specify the time interval between the different measurements of the body weight.
- The method of preparation of the extracted tissues for biochemical analysis should be mentioned.
- I think Echocardiography will add value to the present study because sometimes, ECG changes may be minimal especially in studies of short duration like the present study.
- In Biochemical and antioxidant analysis line 3: the sentence “in the serum of…… using a semi-analyser” is not correct. I think there is a missed part. Please revise.
- The kits used for determination of CRP, CK-MB and LDH should be mentioned.
- I suggest to measure a more specific markers for the cardiac functions such as troponin T and Pro-BNP.
- A reference for “RNA isolation and qPCR analysis” should be added.
- I suggest to investigate the anti-inflammatory and the apoptosis regulating properties of propionic acid in the present study as this will add a great value to the present research.
- The histopathological examination is not sufficient. I suggest to carry out immunohistochemical and electron microscopic study of the cardiac tissues because the changes in these examinations precede the gross changes in H&E sections.
- In statistical analysis, what is meant by “SD and SPSS”?
- Results and discussion:
- I prefer to separate the Results and Discussion sections to best describe and discuss the obtained data.
- A legend for figure 2 should be added.
- Figure 2 and 4: The quality of these figures should be improved.
- Table 1: The meaning of the alphabets “a, b, c, d” should be clearly mentioned.
- In antioxidant assays, MDA was designated as “serum MDA” although it was mentioned in the “Methods” section that it was measured in the cardiac tissues. Please, revise.
- In antioxidant assays, the authors mentioned that “CNPs and DOX+PA groups showed very low level of serum MDA compared to the untreated control” and this wasn’t true. Please, revise.
- The histopathological figures need a detailed legend specifying the pathological events that occur in each group. Also, scale bars should be added to these figures. Also, examination of the aorta needs a higher magnification.
- The discussion should concentrate on analysis of the results of the present study.
- Conclusion:
- I think that the conclusion was not sufficient. The possible clinical implications of the results of the present study should be addressed.
- General comments:
- The manuscript should be revised by English-naïve speaker to improve the quality of the language.
- The manuscript should be checked regarding the grammatical and typing errors and plagiarism.
- The references should be updated.
Author Response
Reviewer Comments 1
- Abstract:
Comment: The meaning of the abbreviations should be clearly defined at their first mention (e.g. UV in line 6 of the abstract).
Reply: Abbreviations have been mentioned correctly throughout the manuscript.
Comment: Line 13: Peroxisome proliferator-activated receptor gamma is NOT a specific cardiac marker. It is present in many tissues of the body other than the heart. Please, revise
Reply: The statement “cardiac biomarker” is removed in the manuscript.
Comment: Line 16: “offer” should be replaced with “offered”.
Reply: The word “offer” is replaced as “offered”
Comment: Key words: The meaning of “PPARγ” should be clearly mentioned.
Reply: The meaning of “PPARγ” is mentioned clearly.
- Introduction:
- Comment: The novel points in this study should be clarified because there are previous studies that demonstrated that propionic acidemia may lead to cardiomyopathy and this is against the hypothesis of the current study.
Reply: Propionic acid is a short chain fatty acid, which is naturally synthesized by ruminants and humans. As mentioned by the reviewer, some of the earlier studies reported that propionic acid may lead to cardiomyopathy, but in the conducted study, the selected dosage is very much less as it is naturally synthesized in our body. So the conducted study was not spoiled the hypothesis proposed.
- Comment: The sentence “Similar to CNPs, PA has been already reported for its lipid lowering ability [20].” In the second paragraph should be transferred to the third paragraph.
Reply: The sentence is transferred into third paragraph.
- Comment: The suggested role of propionic acid in protection against DOX-cardiotoxicity should be re-written in a more-detailed manner to explain the hypothesis of the present study and to clarify the proposed mechanisms.
Reply: it is mentioned in the manuscript in the second sentence of third paragraph in introduction.
- Comment: The reference number [20] used for delineation of the role of propionic acid is old (2010). Please, use updated references.
Reply: The above mentioned article is a seminal article for this conducted study and there are no strong evidence reported elsewhere. Hence, we request the reviewer and editor to allow this reference in the manuscript.
- Comment: The meaning of the abbreviations should be clearly defined at their first mention (e.g. PPARɤ).
Reply: PPARɤ is abbreviated clearly in abstract section.
- Materials and methods:
- Comment: The exact source, concentrations and the catalogue numbers of the used drugs, kits and chemicals should be mentioned.
Reply: it is mentioned elaborately in 2.1 Materials and Methods session
- Comment: A reference for the equation used for determination of “Drug loading efficacy” should be added.
Reply: The reference for the equation is mentioned in 2.3 Drug loading efficacy i.e., Arulmozhi et al [22].
- Comment: A reference for the methods used for “characterization studies” should be added.
Reply: references are added in the manuscript.
- Comment: How did you know that the animals were acclimatized?
Reply: Before conducting in vivo experiment, we generally allow the animals for 2 weeks to acclimatize in the lab environment. Accordingly, we adopted the procedure.
- Comment: The name of the institution that gave the Ethical approval should be mentioned.
Reply: The name of the institution is mentioned in 2.5 Experimental animals
- Comment: The number of animals used in this study should be mentioned.
Reply: Totally 35 animals were used in this study and it is mentioned in 2.6 Experimental design
- Comment: In experimental design line 3: Replace “Adriamycin” with “Doxorubicin”.
Reply: The word Adriamycin is replaced as Doxorubicin.
- Comment: References for the used doses of drugs and duration should be added.
Reply: Reference is added in the manuscript.
- Comment: How the doses were split?
Reply: The doses were split equally from the cumulative dose.
- Comment: The frequency of administration of DOX should be clarified.
Reply: The DOX has administered daily at very low dose to attain the cumulative dose at the end of month.
- Comment: “The body weight was monitored regularly during the experimental period” Please specify the time interval between the different measurements of the body weight.
Reply: The body weight was monitored daily and it is now mentioned in the manuscript.
- Comment: The method of preparation of the extracted tissues for biochemical analysis should be mentioned.
Reply: The reference was given in the manuscript.
- Comment: I think Echocardiography will add value to the present study because sometimes, ECG changes may be minimal especially in studies of short duration like the present study.
Reply: Yes, surely we will add this parameter in the upcoming study.
- Comment: In Biochemical and antioxidant analysis line 3: the sentence “in the serum of…… using a semi-analyser” is not correct. I think there is a missed part. Please revise.
Reply: The above mentioned sentence is corrected as “in the serum of experimental animals using a semi-analyser”
- Comment: The kits used for determination of CRP, CK-MB and LDH should be mentioned.
Reply: The kits used for the determination of CRP, CK-MB and LDH is mentioned elaborately in 2.1 Materials
- Comment: I suggest to measure a more specific markers for the cardiac functions such as troponin T and Pro-BNP.
Reply: In this study, various cardiac specific markers have been analyzed which help to evaluate the functioning of heart. In the upcoming studies we will incorporate the above mentioned specific markers.
- Comment: A reference for “RNA isolation and qPCR analysis” should be added.
Reply: The reference is added in the manuscript.
- Comment: I suggest to investigate the anti-inflammatory and the apoptosis regulating properties of propionic acid in the present study as this will add a great value to the present research.
Reply: In continuation of this study, we are planning to conduct the apoptotic study in near future.
- Comment: The histopathological examination is not sufficient. I suggest to carry out immunohistochemical and electron microscopic study of the cardiac tissues because the changes in these examinations precede the gross changes in H&E sections.
Reply: In the present study, we are tried to emphasis the histological activity based on the results obtained.
- Comment: In statistical analysis, what is meant by “SD and SPSS”?
Reply: SD means Standard Deviation and SPSS means Statistical Package for Social Science
- Results and discussion:
- Comment: I prefer to separate the Results and Discussion sections to best describe and discuss the obtained data.
Reply: In this study, we are comparing 7 different groups and different parameters, so we have written the results and discussion together.
- Comment: A legend for figure 2 should be added.
Reply: The legend of figure 2 is added
- Comment: Figure 2 and 4: The quality of these figures should be improved.
Reply: Figures 2 and 4 are improved
- Comment: Table 1: The meaning of the alphabets “a, b, c, d” should be clearly mentioned.
Reply: The alphabets are clearly mentioned in the table legend clearly.
- Comment: In antioxidant assays, MDA was designated as “serum MDA” although it was mentioned in the “Methods” section that it was measured in the cardiac tissues. Please, revise.
Reply: It is revised in the manuscript
- Comment: In antioxidant assays, the authors mentioned that “CNPs and DOX+PA groups showed very low level of serum MDA compared to the untreated control” and this wasn’t true. Please, revise.
Reply: it is revised in the manuscript
- Comment: The histopathological figures need a detailed legend specifying the pathological events that occur in each group. Also, scale bars should be added to these figures. Also, examination of the aorta needs a higher magnification.
Reply: It is clearly mentioned in the figure legend.
- Comment: The discussion should concentrate on analysis of the results of the present study.
Reply: it was improvised.
- Conclusion:
Comment: - I think that the conclusion was not sufficient. The possible clinical implications of the results of the present study should be addressed.
Reply: It was addressed clearer now.
- General comments:
- Comment: The manuscript should be revised by English-naïve speaker to improve the quality of the language.
Reply: The manuscript is completely revised by Sayantan Ghosh (English-native speaker).
- Comment: The manuscript should be checked regarding the grammatical and typing errors and plagiarism.
Reply: Yes, it is Checked.
- Comment: The references should be updated.
Reply: The references are updated.

Reviewer 2 Report
In this paper the authors demonstrate DOX toxic effects on heart and propose the drug conjugation with CNPs and/or supplementation with PA to reduce DOX toxicity.
The study is very interesting but, in my opinion, some issue should be better explained.
- The authors analyzed some biochemical markers and enzymatic antioxidants to evaluate cardiac damage in the different experimental groups. In the results section they only evidenced the positive effects of CNPs and DCNPs but they did not discuss of the high levels of CRP, CK-MB and LDH induced by DCNPs+PA which are comparable to or, as for CRP, significantly higher than those induced by DOX. Moreover, DCNPs+PA are indicated to upregulate PPARγ expression level underlining the positive effect of this treatment in reducing the DOX cardiotoxicity. How these two different findings can be explained?
- Discussing about the antioxidants assays, the authors stated that “Rats in DCNPs exhibited lower serum level of MDA compared to DOX-administered group, whereas CNPs and DOX+PA groups showed very low level of serum MDA compared to untreated control and other experimental animals”. In Table 1, however, MDA levels in DOX+PA group are 2 fold that of untreated rats. Is there an error? Please, can you explain?
- As histological observation concern, The authors should show an image of longitudinal section of heart tissue for DCNPs+PA as shown for the other groups.
- In qPCR analysis paragraph, the ADRIM+PA group is reported (which should correspond to DOX+PA group) but it is not indicated elsewhere in the paper. Please correct or explain.
- Fig. 2 legend lacks and the figure quality should be improved (it's blurry).
- Nanomolar concentration should be indicated as nM: please correct where needed.
Author Response
Reviewer Comments 2
Comments and Suggestions for Authors
In this paper the authors demonstrate DOX toxic effects on heart and propose the drug conjugation with CNPs and/or supplementation with PA to reduce DOX toxicity.
The study is very interesting but, in my opinion, some issue should be better explained.
- Comment: The authors analyzed some biochemical markers and enzymatic antioxidants to evaluate cardiac damage in the different experimental groups. In the results section they only evidenced the positive effects of CNPs and DCNPs but they did not discuss of the high levels of CRP, CK-MB and LDH induced by DCNPs+PA which are comparable to or, as for CRP, significantly higher than those induced by DOX. Moreover, DCNPs+PA are indicated to upregulate PPARγ expression level underlining the positive effect of this treatment in reducing the DOX cardiotoxicity. How these two different findings can be explained?
Reply: Administration of DCNPs+PA increases the level of MDA and LDL. It may be caused due to the combination of all the drugs. It is mentioned in the result and discussion.
- Comment: Discussing about the antioxidants assays, the authors stated that “Rats in DCNPs exhibited lower serum level of MDA compared to DOX-administered group, whereas CNPs and DOX+PA groups showed very low level of serum MDA compared to untreated control and other experimental animals”. In Table 1, however, MDA levels in DOX+PA group are 2 fold that of untreated rats. Is there an error? Please, can you explain?
Reply: Yes, it is a big mistake and it was typed as typographical error and it is corrected.
- Comment: As histological observation concern, The authors should show an image of longitudinal section of heart tissue for DCNPs+PA as shown for the other groups.
Reply: In this manuscript, the same sectioning was followed in DCNPs+PA group like other tested groups and the damages shown in DCNPs+PA group may be due to degenerative changes.
- Comment: In qPCR analysis paragraph, the ADRIM+PA group is reported (which should correspond to DOX+PA group) but it is not indicated elsewhere in the paper. Please correct or explain.
Reply: It was corrected in the manuscript
- Comment: Fig. 2 legend lacks and the figure quality should be improved (it's blurry).
Reply: It is improved now
- Comment: Nanomolar concentration should be indicated as nM: please correct where needed.
Reply: It is corrected in the manuscript.
Round 2
Reviewer 1 Report
The authors had appropriately addressed all my comments.